# Effects of time management interventions on mental health and wellbeing factors: A protocol for a systematic review

**Anna Navin Young**[ID]*, **Aoife Bourke**[☯]**, Sarah Foley**[☯]**, Zelda Di Blasi**[☯]

School of Applied Psychology, University College Cork, Cork, Ireland

☯ These authors contributed equally to this work.
* anavinyoung@ucc.ie

**Data Availability Statement:** No datasets were generated or analysed during the current study. All relevant data from this study will be made available upon study completion.

## Abstract

### Background

Poor employee mental health and wellbeing are highly prevalent and costly. Time-related factors such as work intensification and perceptions of time poverty or pressure pose risks to employee health and wellbeing. While reviews suggest that there are positive associations between time management behavior and wellbeing, there is limited rigorous and systematic research examining the effectiveness of time management interventions on wellbeing in the workplace. A thorough review is needed to synthesize time management interventions and their effectiveness to promote employee mental health and wellbeing.

### Method

A systematic search will be conducted using the following databases: PsychINFO via OVID (1806-Present), Web of Science, Scopus via Elsevier (1976-Present), Academic Search Complete (EBSCO), Cochrane Library via Wiley (1992-Present), and MEDLINE via OVID (1946-Present). The review will include experimental and quasi-experimental studies that evaluate the effects of time management interventions on wellbeing outcomes on healthy adults in a workplace context. Only studies in English will be included. Two authors will independently perform the literature search, record screening, data extraction, and quality assessment of each study included in the systematic review and meta-analysis. Data will be critically appraised using the Cochrane risk-of-bias tools. Depending on the data, a meta-analysis or a narrative synthesis will be conducted. The Preferred Reporting Items for Systematic Reviews and Meta-Analyses (PRISMA) guidelines were followed in the development of this protocol. The protocol has been registered in PROSPERO (CRD4202125715).

### Discussion

This review will provide systematic evidence on the effects of time management interventions on wellbeing outcomes in the workplace. It will contribute to our understanding of how time management approaches may help to address growing concerns for employee mental health and wellbeing.

**Funding:** The authors received no specific funding for this work.

**Competing interests:** The authors have declared that no competing interests exist.

## Introduction

Each year, the consequences of poor employee mental health and wellbeing cost the global economy an estimated $1 trillion [1]. In 2022, the U.S. Surgeon General raised the issue of workplace wellbeing to national prominence [2]. Time is a critical factor to consider in understanding the current mental health and wellbeing challenges observed in the workplace. In the European Union's 2022 Occupational Safety and Health survey, nearly half of respondents reported that severe time pressure and work overload contributed to increased work stress [3]. Research over the last few decades indicates that work intensification, referring to both the increased pace and increased amount of work, impairs employee wellbeing, health, and motivation [4–6].

Additionally, research on time poverty, or the perception of not having enough time, finds this temporal perception is detrimental to self-assessed mental health and health, emotional wellbeing, work-family conflict, physical activity, life satisfaction, perceived work performance, concentration at work, and turnover intentions [7–13]. Time poverty can also increase stress and stress-related symptoms including headaches, sleep disturbances, and musculoskeletal pains [9, 11, 14].

### Time management interventions

Time management interventions are the most common time-focused interventions implemented in the workplace and may support employee mental health and wellbeing by addressing experiences and impacts of time poverty and work intensification. Definitions of time management vary across the literature, often including components related to goal and priority setting, planning, structuring, organizing, and evaluation [15–19]. Time management interventions consequently vary depending on which definition of time management has been adopted [15, 16].

There is currently some evidence to suggest that time management interventions can improve wellbeing, however there are limitations with this research [15–17]. For example, a non-systematic review identified 35 time management studies using self-report questionnaires, diaries, and experiments published between 1954 and 2005 [15]. The authors reported that time management was positively related to perceived control of time, job satisfaction, and health, and negatively related to factors such as emotional exhaustion, role overload, and work-family conflict. This review identified several methodological limitations within the time management literature. First, the majority of study participants were university students, limiting the results' relevance in a workplace context [15]. Second, a variety of time management definitions were used across studies, with some studies not providing any definition. Further, ten different self-report questionnaires were used to measure time management behaviors. The lack of transparent and consistent operationalization indicates strong heterogeneity, making it difficult to know whether 'time management' is being evaluated consistently across the literature [15]. Third, only eight of the 35 studies evaluated time management interventions, indicating a limited body of experimental research [15]. However, these experiments generally found that time management training increased self-reported time management skills and academic and job performance.

A recent comprehensive meta-analysis of 158 studies (n = 53,957) found time management (assessed based on studies using a quantitative measure of time management) to increase wellbeing, particularly life satisfaction, more than academic and job performance [16]. This meta-analysis further highlighted the limitations identified in the previous non-systematic literature review. First, a majority of studies used cross-sectional designs, thus limiting the relational conclusions that can be drawn between time management and wellbeing outcomes. Second, a

majority of studies involved university students and time management was significantly less impactful for worker populations compared to student samples [16]. Third, there are limited experimental studies done to evaluate the effectiveness of time management interventions. And, finally, there is a lack of clarity, consistency, and generalizability across what is being conducted as a time management intervention [16].

The meta-analysis addressed the question of whether time management works, revealing that time management may primarily enhance wellbeing opposed to performance [16]. However, the question remains whether time management interventions (and which interventions) work to improve wellbeing. A review and synthesis of the time management intervention literature is needed to understand the current state of the field and further provide foundations for future research, development, and application of consistent, valid, and generalizable time management interventions. This is the focus and contribution of this systematic review.

## Aim of the review

The aim of this proposed review is to synthesize experimental and quasi-experimental studies that evaluated the effectiveness of a time management intervention on wellbeing outcomes among healthy adults in a workplace context. As the need for effective interventions grows alongside rising concern for workplace mental health and wellbeing, this review will contribute to our understanding of whether time management interventions may be integrated into impactful solutions. The proposed review aims to answer the following questions:

1. Do time management interventions improve mental health and wellbeing outcomes among healthy working adults?

2. What are the characteristics of effective time management interventions?

## Objectives

1. The primary objective is to critically synthesize the effectiveness of time management interventions on wellbeing among healthy adults in the workplace.

2. The secondary objective of the review is to investigate the types and characteristics of time management interventions that have been conducted in experimental settings.

3. The final objective is to evaluate the quality of the evidence.

## Methods and analysis

The Preferred Reporting Items for Systematic Reviews and Meta-Analyses Protocols (PRISMA-P) guidelines were adhered to in the development of this protocol [20, 21]. The protocol was registered with the International Prospective Register of Systematic Reviews (PROSPERO; CRD42021257157). The systematic review will be carried out following the PRISMA-P checklist (S1 Table) [22] and the Cochrane Handbook for Systematic Reviews of Interventions guidelines [23].

## Types of studies

The acronym PICO (Population, Intervention, Comparison, Outcomes) guided the inclusion and exclusion criteria for the systematic review (Table 1) [24]. This review will include randomised controlled trials and quasi-experiments (controlled, non-randomised, and pre/post-intervention studies). Non-experimental studies, including literature reviews, case reports,

**Table 1. Eligibility criteria for the systematic review.**

| PICO acronym | Inclusion Criteria | Exclusion Criteria |
|---|---|---|
| P—Participant/ Population | Healthy (non-clinical) adults 18+ years of age, workplace context | <18 years old, clinical populations, non-workplace or educational context |
| I—Intervention | Time management intervention or training | No explicit time management intervention/training |
| C—Control/ Comparison | Control group that does not complete the time management intervention | No control/comparison group |
| O—Outcomes | At least one wellbeing-related outcome, performance outcomes will also be recorded but only in studies that include the wellbeing aspect | Studies that do not report on participant outcomes |
| | | Studies with no measured wellbeing-related outcome |
| Additional Criteria | Randomised controlled trials, quasi-randomised trials, non-randomised trials, and experimental studies | Correlational, cross-sectional, and qualitative studies; literature reviews, and case reports |
| | | Non-English language articles |

qualitative, correlational, and cross-sectional studies, will be excluded from the review. Included articles will be written in the English language.

## Types of participants

The review will include studies that involve healthy (non-clinical) adult participants in an organisational or educational context.

**Patient and public involvement.** As this research is based on previously published data, participants were not directly involved or recruited for this study. Participant consent for publication of this research is not required.

## Types of interventions

Studies will be eligible for inclusion if they evaluated the delivery of a time management intervention aimed at enhancing at least one wellbeing-related outcome. The review will include studies that involved one intervention (single component) or two or more interventions (multicomponent). The intervention must be explicitly referred to as a time management intervention, though the review will not limit study inclusion to a specific definition of time management.

## Types of outcome measures

The primary outcomes will be self-reported wellbeing-related outcomes, including life satisfaction, stress, anxiety, exhaustion, burnout, and depression. Studies will only be included in the review if they reported at least one wellbeing-related psychological outcome measure as assessed pre-intervention and post-intervention.

## Search method

The search strategy will be carried out through six specialized and general electronic databases from inception for this review: Medical Literature Analysis and Retrieval System Online (MEDLINE) via PubMed, PsycInfo, Web of Science, Scopus, Academic Search Complete, and Cochrane Library Central. A range of words related to 'time management' and 'wellbeing' will be searched (Table 2). The search will aim to identify published experimental and quasi-experimental studies that evaluated a time management intervention in relation to at least one

**Table 2. Sample systematic review search strategy.**

| |
|---|
| "Time management" OR "time perception" OR "control of time" OR "time allocation" OR "time affluence" OR "time famine" OR "time pressure\*" OR "quality time" OR "quality work time" OR "time balance" OR "meaningful time" OR "time value\*" OR "time orientation" OR "time perspective\*" OR time-related |
| *AND* |
| work OR employment OR job OR occupation OR training OR program OR activity OR initiative OR efficacy |
| *AND* |
| employee\* OR worker\* OR human\* OR adult\* |
| *AND* |
| enhance\* OR improve\* OR mindfulness OR quality OR values OR wellbeing OR stress OR satisfaction OR "quality of life" OR well-being OR happiness OR flow OR habit\* OR routine\* OR pressure\* OR productivity |
| *AND* |
| RCT OR "randomized controlled trial" OR "randomised controlled trial" OR control OR "controlled trial" OR non-ranomi\* OR quasi-randomi\* |

wellbeing-related outcome. The detailed search strategy was developed by the research team in consultation with a Faculty Librarian. The search will be limited to studies published in the English language. The decision to include only studies published in English results from limited resources and the language constraints of our review team. As the aim of this systematic review is to evaluate rigorous experimental studies unpublished grey literature will not be included. The review will include studies published up until 1 July 2023.

Manual searches of references will be conducted in relevant papers, including the reference lists of any studies assessed for inclusion in the review, in attempts to identify any additional eligible studies. PROSPERO and the Cochrane Library will also be searched for any systematic reviews planned or completed that relate to this review. The reference lists of a recent meta-analysis [16] and a previous time management literature review [15] will also be manually searched.

## Study selection

The first and second authors will independently screen papers, first by title and abstract and then by full text. Data will be extracted using a data extraction form and recorded in a shared spreadsheet. Both the extraction form and spreadsheet have been designed for the purposes of this review. Any conflicts which arise in the screening and extraction stages will be resolved through discussion or further involvement of a third researcher (ZDB). A flow diagram will present a record of study screening following the PRISMA-P guidelines. Excluded studies, and their reason for exclusion, will be documented within the flow diagram.

## Data extraction process

The data extraction form has been designed by ANY to record data from studies during the full-text review stage.

The following information will be included in data extraction:

1. Country of origin, author(s), and year of publication

2. Study method: design (e.g., experimental and quasi-experimental)

3. Sample: number of participants, age, gender, and other demographic characteristics

4. Context: Workplace, educational environment

5. Type of intervention: single or multi-component

6. Delivery form

7. Session duration (number of sessions and duration of each session)

8. Control group(s)

9. Number of participants at follow up and overall retention rates.

10. Mean/SD, p-value, and effect size

11. Outcome measures used

**Missing data.** The authors will attempt to contact study authors in the case of missing or incomplete information. The available data will be analysed as reported should study authors be unavailable.

## Risk of bias assessment

In accordance with the Cochrane Handbook, the Cochrane risk-of-bias tool will be used to assess the methodological quality of the included studies. Randomised controlled trials will be assessed using the Risk of Bias II tool (ROB II), while quasi-experimental and nonrandomised trials will be assessed using the ROBINS I tool. Assessment will include methods of randomisation and intervention allocation. Risk of bias will be independently conducted by the first and second author and inter-rater reliability will be calculated using the kappa coefficient. In the case of disagreements, a discussion with a third reviewer (ZDB) will be used to reach a consensus. Study authors will be contacted in the case of insufficient information. The risk of bias assessments will result in a classification of low risk, some concerns, or high risk.

## Data synthesis

Adhering to Cochrane guidelines [25], ANY will lead the authors' conduction of a narrative synthesis. The authors will address any conflicting interpretations that arise during the narrative synthesis through discussion until a consensus is reached. The narrative synthesis will be structured around the included studies, the types of time management interventions and topics used, and the intervention outcomes. The characteristics and components of included interventions will also be analyzed and reported. Wellbeing outcomes will be reported along with the measures used in each study. The authors will calculate the percentage of studies that included each intervention and outcome element. Overall, the narrative synthesis will integrate these findings to provide a comprehensive overview of the current evidence of the effectiveness of time management interventions on workplace wellbeing. This will involve a summary of what the included studies reveal regarding effective time management intervention structures, topics, modes of delivery, and outcomes.

A limited scope for meta-analysis is anticipated due to the range of outcomes measured, measurement types, and the small number of existing trials. Where studies have used the same intervention, comparator, and outcomes measures, a random-effects meta-analysis will be conducted with the pooled results.

Depending on the data gathered, subgroup analyses may be conducted to examine the effects of the type of intervention (single component or multicomponent) and duration of intervention.

## Discussion

The aim of this study is to provide a comprehensive overview of the factors influencing effectiveness of time management interventions aimed at enhancing mental health and wellbeing, based on the evidence of experimental and quasi-experimental studies.

Considering the rise in mental health and wellbeing issues in the workplace and reported time poverty, and despite the popularity of time management tools, little is known about the effectiveness of time management interventions, and what elements of time management are particularly useful.

Effective time management interventions have the potential to promote mental health and wellbeing. However, the history of time management highlights limited evidence-based, empirically evaluated strategies for enhancing time management in work and educational settings [15–17]. The findings of this review are expected to provide an overview of time management interventions that have been conducted using a robust trial design, and their corresponding wellbeing outcomes.

The review will contribute to evaluating these time management interventions from a health and wellbeing perspective, and provide guidance for HR professionals, leaders, and health professionals regarding the current landscape of evidence-based time management interventions and how they may be adopted to support employee wellbeing. Findings from the systematic review will be synthesized and disseminated for relevant stakeholders to promote evidence-based wellbeing initiatives in the workplace.

## Supporting information

**S1 Table. PRISMA-P checklist.**
(PDF)

## Author Contributions

**Conceptualization:** Anna Navin Young, Aoife Bourke, Sarah Foley, Zelda Di Blasi.

**Data curation:** Anna Navin Young.

**Formal analysis:** Anna Navin Young, Aoife Bourke.

**Investigation:** Anna Navin Young.

**Methodology:** Anna Navin Young, Aoife Bourke, Sarah Foley, Zelda Di Blasi.

**Project administration:** Anna Navin Young.

**Supervision:** Sarah Foley, Zelda Di Blasi.

**Writing – original draft:** Anna Navin Young, Aoife Bourke.

**Writing – review & editing:** Anna Navin Young, Sarah Foley, Zelda Di Blasi.

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
