## [Decision Letter · Decision Letter 0]

5 Feb 2024

PONE-D-23-20913Effects of time management interventions on mental health and wellbeing factors: A protocol for a systematic reviewPLOS ONE

Dear Dr. Young,

Thank you for submitting your manuscript to PLOS ONE. After careful consideration, we feel that it has merit but does not fully meet PLOS ONE’s publication criteria as it currently stands. Therefore, we invite you to submit a revised version of the manuscript that addresses the points raised during the review process.

We look forward to receiving your revised manuscript.

Kind regards,

Collins Atta Poku

Academic Editor

PLOS ONE

Journal Requirements:

Reviewers' comments:

Reviewer's Responses to Questions

**Comments to the Author**

1. Does the manuscript provide a valid rationale for the proposed study, with clearly identified and justified research questions?

Reviewer #1: Yes

Reviewer #2: Yes

2. Is the protocol technically sound and planned in a manner that will lead to a meaningful outcome and allow testing the stated hypotheses?

Reviewer #1: Yes

Reviewer #2: Yes

3. Is the methodology feasible and described in sufficient detail to allow the work to be replicable?

Reviewer #1: Yes

Reviewer #2: Yes

4. Have the authors described where all data underlying the findings will be made available when the study is complete?

Reviewer #1: Yes

Reviewer #2: Yes

5. Is the manuscript presented in an intelligible fashion and written in standard English?

Reviewer #1: Yes

Reviewer #2: Yes

6. Review Comments to the Author

You may also provide optional suggestions and comments to authors that they might find helpful in planning their study.

Reviewer #1: The manuscript is well written with sound methodological considerations for the systematic review. Time management is a challenge for many workers and this tend to affect their mental wellbeing as they are often stressed. Finding effective time management interventions is very critical, and the authors are commended for taking this up. I look forward to the findings of the review.

Kindly note the spelling of the word "sever" in line 56. Not sure if the intended word is severe.

On line 68, letter 's' may be added to the word intervention, the last word in the line.

Best wishes.

Reviewer #2: This is a straightforward protocol and will suggest some few points for your consideration:

1. Will it be helpful if the search terms include more specific psychological well-being outcomes such as anxiety etc? Will it change the search results? Please consider further.

2. The data synthesis approach can be sharpened further. For instance, how many authors will be involved in the process, how will conflicting interpretations be resolved? How will the findings be integrated?

7. PLOS authors have the option to publish the peer review history of their article (what does this mean?). If published, this will include your full peer review and any attached files.

Reviewer #1: No

Reviewer #2: No

---

## [Author Response · Author response to Decision Letter 0]

14 Feb 2024

PONE-D-23-20913

Effects of time management interventions on mental health and wellbeing factors: A protocol for a systematic review

We thank the academic editor and reviewers for their consideration of this manuscript and their thoughtful feedback. Below, we respond to each point raised and how, where applicable, these have influenced revisions made to the manuscript. We believe this feedback and corresponding revisions have served to clarify and strengthen the manuscript and we again thank the academic editor and reviewers for their meaningful contributions to this process.

Journal Requirements:

- We apologize for our formatting errors. We have reviewed the manuscript to adhere to formatting guidelines. The tables have been removed as separate file uploads and are only presented within the main manuscript. Supporting information is presented as “S1 Table” in the main manuscript and provided as a separate file upload with correct file naming. 

- We have reviewed our reference list and confirm that it is complete, correct, and abides by the journal’s formatting guidelines. We have checked all references in the Retraction Watch Database and do not reference any retracted papers in our manuscript.

Reviewer #1: 

The manuscript is well written with sound methodological considerations for the systematic review. Time management is a challenge for many workers and this tend to affect their mental wellbeing as they are often stressed. Finding effective time management interventions is very critical, and the authors are commended for taking this up. I look forward to the findings of the review.

- Thank you very much for this feedback. We are pleased to hear that you resonate with the potential contributions of this work.

Kindly note the spelling of the word "sever" in line 56. Not sure if the intended word is severe.

On line 68, letter 's' may be added to the word intervention, the last word in the line.

- These are very detailed catches. Thank you for bringing them to our attention. We have updated the manuscript to reflect these changes.

Reviewer #2:

This is a straightforward protocol and will suggest some few points for your consideration:

1. Will it be helpful if the search terms include more specific psychological well-being outcomes such as anxiety etc? Will it change the search results? Please consider further.

- Thank you for this consideration. Due to the systematic review’s aim of assessing nonclinical populations (in workplace contexts being exposed to time management interventions), we decided to exclude search terms such as anxiety and depression. 

In response to your feedback, we have reviewed the studies included in Aeon et al.’s (2021) extensive meta-analysis of time management. Studies that measured well-being variables (including factors such as anxiety, depression, and psychological distress) were reviewed to see whether they would fit our inclusion criteria. As Aeon et al.’s (2021) meta-analysis should be a comprehensive review of the time management literature, this brief review of their wellbeing-related studies suggests that we have not missed any studies relevant to our review by excluding terms such as anxiety. As a result, we have decided to maintain the original search terms.

2. The data synthesis approach can be sharpened further. For instance, how many authors will be involved in the process, how will conflicting interpretations be resolved? How will the findings be integrated?

- Thank you for this feedback. We have provided additional details in the data synthesis section to address these concerns. Details regarding how and which authors will be involved in data synthesis, how conflicting interpretations will be addressed, and how the findings will be integrated have all been included.

---

## [Decision Letter · Decision Letter 1]

27 Feb 2024

Effects of time management interventions on mental health and wellbeing factors: A protocol for a systematic review

PONE-D-23-20913R1

Dear Dr. Young,

We’re pleased to inform you that your manuscript has been judged scientifically suitable for publication and will be formally accepted for publication once it meets all outstanding technical requirements.

Kind regards,

Collins Atta Poku

Academic Editor

PLOS ONE

Additional Editor Comments (optional):

Reviewers' comments:

Reviewer's Responses to Questions

**Comments to the Author**

1. Does the manuscript provide a valid rationale for the proposed study, with clearly identified and justified research questions?

Reviewer #2: Yes

2. Is the protocol technically sound and planned in a manner that will lead to a meaningful outcome and allow testing the stated hypotheses?

Reviewer #2: Yes

3. Is the methodology feasible and described in sufficient detail to allow the work to be replicable?

Reviewer #2: Yes

4. Have the authors described where all data underlying the findings will be made available when the study is complete?

Reviewer #2: Yes

5. Is the manuscript presented in an intelligible fashion and written in standard English?

Reviewer #2: Yes

6. Review Comments to the Author

You may also provide optional suggestions and comments to authors that they might find helpful in planning their study.

Reviewer #2: Many thanks to the authors for addressing the comments raised. Looking forward to reading this paper when published.

7. PLOS authors have the option to publish the peer review history of their article (what does this mean?). If published, this will include your full peer review and any attached files.

Reviewer #2: No

---

## [Editor Report · Acceptance letter]

1 Mar 2024

PONE-D-23-20913R1 

PLOS ONE

Dear Dr. Young, 

I'm pleased to inform you that your manuscript has been deemed suitable for publication in PLOS ONE. Congratulations! Your manuscript is now being handed over to our production team.

Kind regards, 

on behalf of

Dr. Collins Atta Poku 

Academic Editor

PLOS ONE